# Challenges to diagnosing and managing preeclampsia in a low-resource setting: A qualitative study of obstetric provider perspectives from Ghana

Namratha Atluri[1]*, Titus K. Beyuo[2], Samuel A. Oppong[2], Cheryl A. Moyer[3,4], Emma R. Lawrence[3]

1 University of Michigan Medical School, Ann Arbor, Michigan, United States of America, 2 Department of Obstetrics and Gynecology, University of Ghana Medical School, Korle Bu, Accra, Ghana, 3 Department of Obstetrics and Gynecology, University of Michigan, Ann Arbor, Michigan, United States of America, 4 Department of Learning Health Sciences, University of Michigan, Ann Arbor, Michigan, United States of America

* drbeyuo@gmail.com

**Data Availability Statement:** All relevant data are within the paper and its Supporting Information files.

## Abstract

Preeclampsia is a leading cause of global maternal morbidity and mortality. The greatest burden of disease is in low- and middle-income countries where healthcare providers face significant, understudied, challenges to diagnosing and managing preeclampsia. This qualitative study used semi-structured interviews to explore the challenges of diagnosing and managing preeclampsia from the perspectives of obstetric doctors. Participants were doctors who provide obstetric care at the Korle Bu Teaching Hospital, an urban tertiary hospital in Ghana. Purposive sampling identified doctors with meaningful experience in managing patients with preeclampsia. Thematic saturation of data was used to determine sample size. Interviews were audio recorded, transcribed verbatim, coded using an iteratively-developed codebook, and thematically analyzed. Interviews were conducted with 22 participants, consisting of 4 house officers, 6 junior obstetrics/gynecology residents, 8 senior obstetrics/gynecology residents, and 4 obstetrics/gynecology consultants. Doctors identified critical challenges faced at the patient, provider, and systems levels in detecting and managing preeclampsia, each of which mediates the health outcomes of a pregnancy complicated by preeclampsia. Challenges centered around three overarching global themes: (1) low education levels and health literacy among women, (2) insufficient number of healthcare providers highly trained in obstetric care, and (3) inadequate health infrastructure to support critically ill patients with preeclampsia. Recognizing and addressing root challenges to preeclampsia care has great potential to improve outcomes in pregnancies complicated by preeclampsia in low-resource settings.

**Funding:** Funding sources include the North Pacific Global Health Fellowship (E.R.L), NIH T-35 Short-Term Training Grant for Medical Students (N.A), and the University of Michigan Woll Scholarship through Global REACH (N.A). The funders had no role in study design, data collection and analysis, decision to publish, or preparation of the manuscript.

**Competing interests:** The authors have declared that no competing interests exist.

## Introduction

Preeclampsia is a leading cause of global maternal morbidity and mortality, complicating between 2% and 8% of pregnancies worldwide [1, 2]. In addition to causing immediate complications such as acute kidney injury, liver damage, seizures, stroke, and coagulopathy, preeclampsia is associated with the development of subsequent chronic conditions including essential hypertension and cardiovascular disease [1, 3, 4]. In the past three decades, the incidence of preeclampsia has increased by 11% globally, and it has become the single leading cause of maternal mortality in many low-resource settings [2, 5–7]. This increase can potentially be explained by the rising prevalence of risk factors such as extremes of reproductive age, multiple gestation, and medical comorbidities such as chronic hypertension, obesity, and diabetes.

Preeclampsia exists along a spectrum of hypertensive disorders of pregnancy (HDP), which includes gestational hypertension, preeclampsia without severe features, preeclampsia with severe features, and eclampsia [3]. It can develop any time after 20 weeks of gestation and up to 6 weeks postpartum [3, 4]. Preeclampsia without severe features is diagnosed based on a combination of elevated blood pressure values and proteinuria while preeclampsia with severe features is diagnosed when symptoms are present or specific thresholds for blood pressure elevation ($>/ = 160/100$) or laboratory values (elevations in creatinine, elevations in liver function tests, decreases in platelets) are present [3]. This distinction between clinical diagnoses is essential as it in turn drives management recommendations [3]. Management of preeclampsia depends on severity and requires balancing neonatal risk of preterm delivery with maternal risks of prolonging pregnancy [8].

Globally, rates of progression from preeclampsia to eclampsia have significantly declined due in part to increasing utilization of antenatal care, higher rates of in-facility delivery and greater access to magnesium sulfate [6, 9, 10]. However, these generalized improvements in management have not necessarily translated into better clinical outcomes in low-and middle-income countries (LMICs). Over 99% of maternal deaths due to hypertensive disorders of pregnancy occur in LMICs with a higher incidence of progression to eclampsia and nearly 50% of eclamptic seizures occurring outside of a hospital setting [2, 11]. In Ghana where this study is proposed, incidence of hypertensive disorders in pregnancy is estimated at 7.6%. Despite improvements in clinical protocols and increased access to key medications like antihypertensives and magnesium sulfate, rates of eclampsia continue to be significantly higher than in high-income countries, and neonatal outcomes are poor. Institutional reports from two major tertiary-level hospitals suggest that hypertensive disorders have overtaken hemorrhage as the leading cause of maternal mortality [7, 8]. Unique challenges to management of preeclampsia in LMICs contribute to these disparities, including low levels of health literacy, poor antenatal care attendance, and scarcity of healthcare resources [11–14].

This study aims to understand the challenges of diagnosing and managing preeclampsia in a low-resource setting from the perspectives of obstetric doctors who work at a tertiary level hospital in urban Ghana. With significant patient care experiences spanning a variety of clinical training sites, obstetric doctors are well-positioned to comment about the daily challenges from multiple levels (patient-level, provider-level, and systems-level) they face in providing care to patients with preeclampsia.

## Methods

### Design

This was a qualitative study consisting of semi-structured interviews with obstetric providers, focusing on providers' experiences caring for pregnant patients with preeclampsia, the current state of practice, and challenges faced.

### Setting

This study took place at the Korle Bu Teaching Hospital (KBTH), Ghana's largest tertiary level hospital. The KBTH Department of Obstetrics and Gynecology (OBGYN) runs a six-floor maternity unit with 275 inpatient beds, a perinatal assessment unit, a dedicated high-risk unit, labor and delivery wards, and an outpatient department for gynecology and obstetrics clinic visits. Of the approximately 10,000 deliveries that occur annually at KBTH, 15% are complicated by HDP, the leading cause of maternal mortality at KBTH [7]. HDP is often detected incidentally with no symptoms at routine ANC or with moderate to severe symptomatology as referrals/ emergency transfers from other hospitals.

### Clinical context

At KBTH, patients with HDP are managed according to international clinical recommendations, with adjustments made to account for limited systems for emergency obstetric care [3, 6]. In cases of well-controlled gestational hypertension and preeclampsia without severe features, pregnancies can be closely monitored until 37 weeks gestation, at which time delivery is recommended [3]. Notably, safe monitoring requires serial assessments of blood pressure, laboratory values, and fetal status [15]. While this is accomplished with frequent antenatal visits in high-income settings, the current protocol at KBTH (and in hospitals throughout Ghana) recommends admission of preeclamptic patients to the hospital until delivery. Given the high risk of complications, pregnancies with preeclampsia with severe features and eclampsia are managed with stabilization and delivery [3]. This is a particularly difficult scenario when diagnoses are made at a preterm, peri-viable or pre-viable gestational age, with delivery leading to prolonged Neonatal Intensive Care Unit (NICU) admissions, neonatal morbidity, or stillbirth [8]. Additional management principles include inpatient monitoring until blood pressures stabilize postpartum, administration of antihypertensive medications to control elevated blood pressures, and administration of magnesium sulfate to decrease the risk of maternal seizures [3, 4, 9, 14–16].

### Participants

Participants were doctors who provide care to pregnant women, work primarily at KBTH, and have experience diagnosing and/or managing patients with preeclampsia. All participants were either (1) house officers (recent medical school graduates rotating through core clinical services), (2) OBGYN residents (trainees in obstetrics and gynecology who have completed house officer training), and (3) OBGYN consultants or "attendings" (doctors who have completed specialty training in obstetrics & gynecology).

**Recruitment.** A list of eligible participants was compiled using the OBGYN departmental roster at KBTH, which listed every doctor and their rank (total 152 participants eligible). With guidance from a local research team member at KBTH, purposive sampling was used to identify doctors with meaningful experience diagnosing and managing patients with HDP. Participants were intentionally sampled to achieve representation from each level of clinical experience. After completion, each interview was transcribed and reviewed by the researchers.

Thematic saturation of data was used to decide the final number of participants, with data collection stopping once no new ideas were encountered in the interviews. Interviews were conducted with a total of 22 participants, consisting of 4 house officers, 6 junior residents, 8 senior residents, and 4 consultants (Table 1).

## Procedures

A semi-structured interview guide consisted of a series of broad, open-ended questions with a list of more specific follow-up prompts. Participants' thoughts and opinions were gathered while attempting to limit the bias of researchers' assumptions about preeclampsia care in Ghana. The researchers' bias was limited by purposefully creating a semi-structured interview guide with very broad, open-ended questions to encourage participants to share any thoughts or opinions that come to their mind. In addition, the primary researcher conducting interviews was intentionally chosen to be someone without prior experience working within the Ghanaian healthcare system. The credibility of participants as doctors providing obstetric care established qualitative trustworthiness.

Interviews were conducted in-person at selected quiet places within the KBTH compound. All interviews were conducted by a non-Ghanaian female medical student to ensure objectivity and limit the impact of preconceived notions. She received training on qualitative interviewing techniques and cultural humility by senior members of the research team who have significant experience with the Ghanaian healthcare system. Interviews lasted approximately 30 minutes

**Table 1. Demographics of interviewed participants.**

| Characteristic | Frequency (Proportion) |
|---|---|
| Clinical role | |
| House Officer | 4 (18.2%) |
| Junior Resident in Obstetrics/Gynaecology | 6 (27.3%) |
| Senior Resident in Obstetrics/Gynaecology | 8 (36.4%) |
| Consultant ("attending") in Obstetrics/Gynaecology | 4 (18.2%) |
| Gender | |
| Male | 18 (81.8%) |
| Female | 4 (18.2%) |
| Other/ Prefer Not To Respond | 0 (0%) |
| Years in practice as a doctor | |
| <1 | 4 (18.2%) |
| 1–5 | 2 (9.1%) |
| 6–10 | 8 (36.4%) |
| 11–20 | 7 (31.8%) |
| >20 | 1 (4.5%) |
| Average patients with preeclampsia managed **weekly** | |
| 0–5 | 5 (22.7.0%) |
| 6–10 | 8 (36.4%) |
| 11–15 | 4 (18.2%) |
| 16–20 | 5 (22.7%) |
| Average patients with eclampsia managed **monthly** | |
| 0 | 3 (13.6%) |
| 1 | 7 (31.8%) |
| 2 | 7 (31.8%) |
| 3 | 5 (22.7%) |

and were conducted in January and February 2022. No incentive was offered for participation. Ethical approval was received from KBTH (KBTH-STC 00098/2021) and the University of Michigan (HUM00200589). Prior to starting interviews, all participants completed written informed consent.

## Analysis

Demographic data were descriptively analyzed using frequencies and proportions. Interviews were audio recorded and transcribed verbatim. Transcripts were uploaded into NVivo 12.0 for organization of qualitative analysis. Transcripts were independently reviewed by two researchers (primary interviewer and senior research team member) who separately developed sets of key-word phrases. Across meetings spanning several weeks, an iterative process was used to compare key-word phrases and develop a progressive collective list of codes. Once this list of codes stabilized, a codebook was formally developed consisting of a list of codes with definitions and inclusion and exclusion guidelines. Transcripts were then coded in full. With the involvement of a third senior researcher, the coded transcripts were thematically analyzed using the Attride-Sterling Framework for qualitative analysis, through which a system of global themes and sub-themes was developed [17].

## Results

Doctors highlight key challenges to diagnosis and management that contribute to poor outcomes for many women with preeclampsia. Challenges were conceptualized as patient-level, provider-level, and systems-level, with each level of challenges mediating the relationship between a pregnancy complicated by preeclampsia and subsequent clinical outcomes (Fig 1).

### Patient-level challenges to preeclampsia care

Physicians consistently reported patient-level challenges to be the most significant in providing preeclampsia care (Table 2). Low levels of health literacy amongst patients, in the context of low formal education, was thought to be the root cause of most patient-level challenges.

Many Ghanaian women believe that pregnancy is a safe and natural process and are not aware of pregnancy complications such as preeclampsia or the associated risks. Some doctors noted that even women who are currently being managed for a diagnosis of preeclampsia, or had preeclampsia in prior pregnancies, may not actually understand their diagnosis.

> *"Even those with preeclampsia, some of them don't know. Like you will be managing them and if another person asks them, they don't even understand what you're managing them for." ID 3 HO Female*

Due to this lack of awareness, women are less likely to seek medical care when they experience warning signs or symptoms of preeclampsia. Because many women are not exposed to public health and clinical education about pregnancy, they often rely on socially-spread misinformation and pregnancy myths. Patients' religious and cultural beliefs may contradict medical recommendations (such as declining cesarean delivery as the recommended mode of delivery) or prevent them from seeking healthcare earlier (due to viewing symptoms not as illness).

> *"It takes a lot of time to teach [patients] basic things, like if you are getting swollen, edema, it may be a sign [of preeclampsia]. Because traditionally in some areas, they might think that when you are getting edema it may be an indication that the fetus is a male." ID 20 C Male*

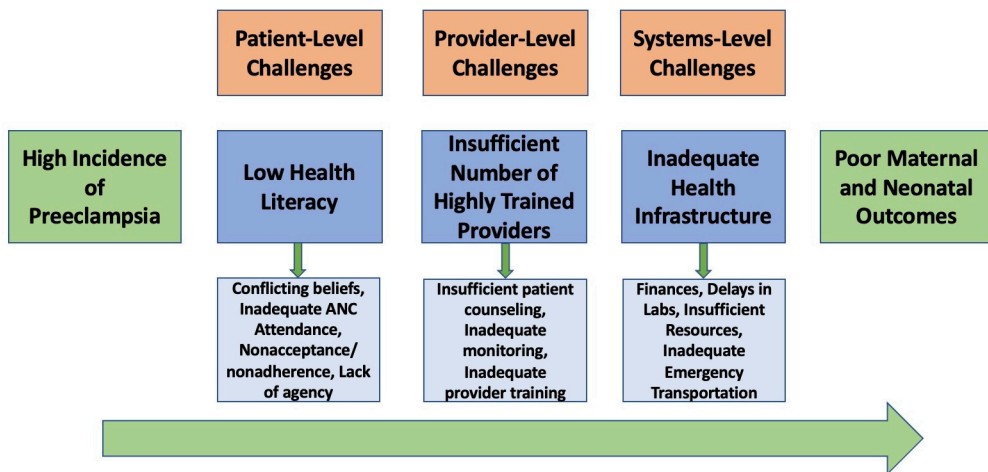

**Fig 1. Patient-level, provider-level, and systems-level barriers to preeclampsia diagnosis and management.**

This low health literacy then contributes to poor ANC attendance, which providers described as one of the biggest challenges to detecting preeclampsia early before complications develop. Many women do not perceive any value in attending ANC, especially if they are asymptomatic or their relatives/ friends have had uneventful pregnancies in the past. This is compounded by long queues to be seen at ANC, where patients may have to wait more than five hours to be seen, as well as competing social and familial responsibilities at home that may prevent them from spending long hours at ANC.

**Table 2. Patient-level challenges to preeclampsia diagnosis and management.**

| Challenges | Representative Quotation |
|---|---|
| **Low health literacy/formal education** | "Another thing is that education. Some people really don't even know that they can have lots of complications that just arise because you are pregnant. They don't know you can have preeclampsia." ID 5 JR Male |
| **Conflicting cultural/ religious beliefs** | "Maybe if you decide at the end of the day [that the] recommended mode of delivery would be a cesarean, they will say no. 'My pastor says if a knife touches me, I will die'. And they will not agree." ID 14 JR Male |
| **Inadequate antenatal care attendance** | "The challenge of nonattendance is a big problem. I mean there are quite a number of women who for one reason or another don't go for antenatal clinics. So, once you are not going for antenatal clinics, you're not checking your BP. The disease is by and large asymptomatic before things get worse, [so] definitely it's going to be difficult to pick it up" ID 7 JR Male |
| **Non-acceptance of diagnosis & non-adherence with medical recommendations** | "Patient accepting [preeclampsia] is one of the challenges. And then [the] patient accepting that their management may need for them to make some sacrifices. And for intensive monitoring, they may need [admission] for a few days. It's very challenging. And then when it comes to those whose condition is even worse off, needing termination of [pregnancy], it's very unacceptable to a majority of our women. "ID 6 SR Male |
| **Lack of agency to make healthcare decisions** | "Some of them are not able to make decisions for themselves. It's all about the empowerment. The husband will have to say come [to the hospital] or stay at home." ID 13 JR Male |

*"Some [patients] think that when they come, we don't do anything for them. We are going to give them the same hematinics, same folic acid, so why not just buy everything from the pharmacy?"* ID 10 HO Male

Even when patients do seek medical care, providers report that they experience challenges with patients accepting a new diagnosis of HDP/ preeclampsia. Elevated blood pressures are often attributed to stress at home, anxiety traveling to and waiting at ANC, or fear of doctors or BP machines, even when evidence of multiple elevated blood pressures, symptoms, or abnormal proteinuria is present. In addition, providers report that pregnant women often don't adhere to management recommendations. Women may refuse hospital admission for blood pressure monitoring due to not liking the hospital environment and not wanting to miss work or be away from family for a long time.

*"They don't understand the need for them to be monitored on admission. They feel alright. My blood pressure is high, but I don't have any symptoms. Why do you want to keep me here? So that's a challenge."* ID 18 SR Male

Within the context of a patriarchal Ghanaian society, formal education is often not equally prioritized for women, especially those coming from lower socioeconomic homes. This can not only lead to the low health literacy that interferes with care-seeking behavior, but also to financial dependence on male partners. As a result, some women may lack autonomy to make their own healthcare decisions and often defer to their male partners.

Despite the many patient-level challenges to providing preeclampsia care, doctors generally still believe that their patients want to be involved in their own healthcare and are motivated to take good care of themselves and their babies.

*"Most of the times, they care about themselves and the baby. So when you ask them to do something, and they know that it's for the benefit of the baby, they will want to. Unless they cannot.."* ID 3 HO Female

**Table 3. Provider-level challenges to preeclampsia diagnosis and management.**

| Challenges | Representative Quotation |
|---|---|
| **Insufficient number of highly-trained healthcare workers** | "The problem boils down to adequate and appropriate staffing. You may have a healthcare facility that has close to 30 patients with probably only one midwife or one healthcare worker. And that's a mismatch. So you'll definitely have errors there with regards to identifying preeclampsia." ID 22 C Male |
| ↓ ↓ ↓ | |
| **Insufficient time for Patient Counseling** | "If the person has preeclampsia, as a doctor you manage [them] right. But the issue with the high number of patients and the smaller number of doctors is that there's not adequate time to sit down with each patient and explain in detail about the condition, the features and all that." ID 13 JR Male |
| **Inadequate Monitoring** | "Our patient to nurse ratio; there's a big mismatch. There are three nurses for a big ward such as this, 30 patients in all. It's quite difficult for you to have good monitoring. It's very stressful for our nurses, and we are not doing the best of monitoring that we can. And it is because we are understaffed." ID 6 SR Male |
| **Inadequate provider training/ knowledge** | "But some facilities, when they refer patients they're at 36 weeks with preeclampsia. And you go through the antenatal records, and you realize that maybe around 31 weeks, they had significant pressures with proteins 3+ but nobody [saw] it to be relevant." ID 14 JR Male |

## Provider/ hospital-level challenges to preeclampsia diagnosis & management

While patient-level challenges were deemed the most significant, physicians also recognized provider-level challenges to detecting and managing preeclampsia (Table 3).

The mismatched ratio between a large number of patients and a small number of highly-trained healthcare providers was thought to be the root cause of most provider-level challenges. Notably, doctors did not believe the high patient loads affected their ability to accurately diagnose preeclampsia or provide standard-of-care clinical management, including administration of antihypertensives or magnesium sulfate. However, the high ratio often affects their ability to counsel their patients adequately regarding preeclampsia and its risks, potentially contributing to noncompliance with management recommendations.

> *"We don't provide a lot of counseling because we have lots of patients. So it's just a very quick thing. Oh your BP is high, take your BP medications. So [patients] may still go out and still not have good understanding and still not take their medications and all." ID 16 HO Female*

Further, the low number of nurses and midwives who staff the inpatient wards results in challenges with monitoring admitted patients as frequently as recommended, especially since continuous blood pressure monitors are not always available. Doctors emphasized that is particularly detrimental to preeclampsia management where the frequent monitoring of blood pressures and assessment of magnesium sulfate side effects are essential to prevent complications.

Providers stated that in addition to the absolute number of healthcare providers being a limitation, gaps in existing provider preeclampsia knowledge and training are also a problem. While not a big issue at tertiary level hospitals, where patients are seen primarily by obstetricians/ gynecologists, doctors reported that they often see cases referred from lower-level healthcare facilities where elevations in blood pressures were not recognized as important or risk factors were not identified to initiate prophylactic measures. They attributed these failures

**Table 4. System-level challenges to preeclampsia diagnosis and management.**

| Challenges | Representative Quotation |
|---|---|
| **Inadequate healthcare infrastructure** | "If there ends up being difficulty [managing preeclampsia], it's the cost [and] absence of resources needed to make decisions." ID 1 SR Male |
| **Lack of Finances** | "Cost. Even though there's free maternal healthcare in Ghana, it's not still everything that is covered. Especially the laboratory investigations and the medications. You may not get that free." ID 17 SR Female |
| **Delays in Laboratory Results** | "The labs, too, they tend to delay. The ones we run here, it takes hours. Sometime days. So maybe the next day you can actually get the labs." ID 12 JR Male |
| **Insufficient medical supplies, equipment, and resources** | "Healthcare facilities are flooded with automated BP machines that are not either validated or calibrated. So majority of the time you are not 100% sure what the BP recordings we are actually getting in." ID 22 C Male |
| **Inadequate emergency transportation system** | "Our traffic, our ambulance services are not the best. They're not really readily available. If you were to call the ambulance, it would take approximately an hour before you get an ambulance sent." ID 17 SR Female |

to inadequate training and up-to-date knowledge of preeclampsia management guidelines at primary and secondary healthcare facilities, where providers managing pregnant women may not be well-trained in obstetric complications. Additionally, doctors felt the knowledge and training of non-physician ancillary healthcare staff (nurses, midwives, etc.) even at tertiary care hospitals could be better.

> *"The midwife can actually see a patient from onset to delivery without a doctor seeing them at the district level. So to the best of their knowledge, they could overlook [high BP] and feel like it's normalcy. A pregnant woman could come in and say headache and they will start treating for malaria, not knowing that it could actually be preeclampsia." ID 5 JR Male*

### Health system challenges to preeclampsia diagnosis & management

Physicians frequently commented about how inadequate health infrastructure also creates challenges in managing preeclampsia (Table 4).

Due to the multitude of limitations including human resources, the healthcare system in Ghana relies on patients and their families to assist with many aspects of healthcare delivery. This includes tasks like transporting lab samples, picking up results, and buying medications prescribed in the inpatient setting.

> *"Relatives have to chase, literally take the blood samples, send it to the lab, go and pick up the results and all that. If I were to collapse right now, there is no relative here. I am expected to, in my unconscious state, find a relative who will move my blood around and buy my medications. So, it's a major challenge." ID 17 SR Female*

Doctors unanimously reported that the cost of healthcare relative to patient poverty is one of the biggest barriers for preeclampsia management. While pregnancy care is covered under Ghana's National Health Insurance Scheme, and rates of insurance are very high, many aspects of preeclampsia care are not covered or only partially covered by insurance. Point-of-care payments are needed prior to accessing care, requiring patients or relatives to produce cash before medications can be administered or laboratory tests can be collected. This exists in a context of high levels of poverty, with patients often unable to afford small healthcare payments let alone extended admissions for preeclampsia monitoring or management of complications. This affects all aspects of preeclampsia care, from issues affording the cost of transportation to attend frequent ANC as well as follow through with management recommendations including the cost of hospital admission, labs and medications.

> *"I think the main challenge is the long hospital stay and then the financial aspects of staying in the hospital and being managed for preeclampsia. Also cost of labs [and] medication are some of the issues that might also be attributed to the difficulty managing preeclampsia." ID 5 JR Male*

All physicians commented about the systemic inefficiencies that lead to delays in laboratory investigations, which are critical to preeclampsia care. This includes no insurance coverage for many labs, no dedicated healthcare personnel to collect and transport specimens, and long backlogs to process results. Providers reported that these delays in labs results in a lack of clinical information and negatively affects their ability to make good management decisions for

their patients. Despite the unavailability of this crucial information, providers still are responsible for making time-sensitive decisions regarding diagnosis and timing of delivery.

*"Sometimes, we deliver the [patient] because [doctors] are tired of working in a vacuum. We don't know the kidney function, we don't know the liver function, we don't know when the money will come. So let's just deliver this lady now [as] we know delivery is the answer to preeclampsia. As to whether that is the best management for the patient, I am not sure it is." ID 4 C Female*

Even at a tertiary level hospital like KBTH, availability of medical equipment and resources is limited with poorly functioning BP monitors, very few dialysis machines, and low number of ventilators. There is often low bed capacity both in the general wards and Intensive Care Unit. While access to antihypertensives and magnesium sulfate has significantly improved, availability of timely blood products is still a major issue.

*"Blood is always in short supply. If the patient has a coagulation defect, we need cryoprecipitate [and] platelet concentrate. We have to put in a request in the morning, and it will be ready at 5 PM. But, you need it right now." ID 4 C Female*

In addition, physicians reported that the current ambulance system in Ghana is inadequate and leads to delays in receiving care during an emergency. This unreliability of transportation contributes to why current management guidelines recommend admission of patients for monitoring so that treatments are not delayed if clinical status deteriorates.

## Discussion

### Principal findings

Interviews with obstetric doctors at a tertiary care hospital in urban Ghana identified critical challenges faced at the patient, provider, and systemic level to detecting and managing preeclampsia, which mediate the health outcomes of a pregnancy complicated by preeclampsia. All challenges centered around three key overarching global themes: (1) low education levels and health literacy among pregnant women, (2) insufficient number of health providers highly trained in obstetric care, and (3) inadequate health infrastructure.

### Findings in context of literature

While many studies have described disparities in preeclampsia outcomes in LMICs compared to high-income countries (HICs), very limited studies have analyzed the diagnostic and management challenges that contribute to these disparities. The findings of our qualitative study were largely similar to the challenges identified by the few studies examining preeclampsia care in low-resource settings. Similar to the patient-level challenges identified in our study, studies conducted in Ethiopia, Haiti, Zimbabwe, and Nigeria also described how patients' limited understanding of preeclampsia symptoms, predilection for attributing high blood pressures to psychosocial stressors, and sociocultural beliefs about management recommendations interfered with regular ANC attendance, early detection of preeclampsia, and provision of adequate care [18–20]. Our study specifically highlights how the onerous process of attending ANC (may take an entire day), social and familial obligations of women in Ghana, and often low agency also complicates care-seeking behavior.

These studies also supported our findings about provider-level and systems-level challenges with consistent mentions of insufficient availability of providers, medicines, functioning

hospital equipment, and laboratory results hindering the appropriate management of pre-eclampsia [14, 20–22]. Notably, a study examining the adherence to WHO screening and management guidelines for preeclampsia in six sub-Saharan African countries showed extremely low use of recommended practices and availability of magnesium sulfate for acute hospital treatment, despite it being on the essential drug list in all surveyed countries [14]. Magnesium sulfate availability may have been less of a problem in our study since our study site was a tertiary hospital in a capital city. A qualitative study in Ghana also showed inconsistencies in national management guidelines that led to confusion and poor adherence amongst midwives and community health nurses not well-trained in obstetric complications [21]. Our study specifically sheds light on how inadequate patient counseling due to low provider-patient ratios contributes to patients' low health literacy, infrequent ANC attendance, and poor adherence to recommendations. Altogether, these findings highlight how shortcomings in provider training and scarcity of healthcare resources leads to inadequate detection of preeclampsia and subsequent management.

## Healthcare and policy implications

Addressing the key challenges to providing preeclampsia care in low-resource settings, as identified by our qualitative study and supported by other studies [14, 18–22], has great potential to improve maternal and neonatal outcomes. Proposed solutions should center around our study's three overarching global themes.

First, while promotion of formal education for women is the cornerstone to improving health literacy and thereby care-seeking behavior, public health campaigns and focused education during ANC by ancillary healthcare staff can also lead to a better understanding of the risks and complications of preeclampsia and importance of adherence to medications and management recommendations. Even with appropriate health literacy, our study shows that the social and familial responsibilities of women may still hinder their ability to access healthcare. Therefore, innovative care delivery modalities, such as home blood pressure monitoring or telehealth interventions, should also be explored to empower women to become more involved in monitoring their health [22–24].

Second, healthcare worker shortage is not a problem unique to managing preeclampsia or to LMICs, although it is much more common in these settings. Systemic policies allowing for training of more healthcare providers is the long-term solution to this challenge. However, our study shows there is opportunity for better training of existing providers, such as midwives and non-obstetric doctors in primary health facilities, that still may lead to the desired result of improved clinical outcomes [21]. Interventions should focus on creating simple but comprehensive management guidelines and plan workshops with healthcare providers at all levels and settings to disseminate these guidelines.

Finally, inadequate health infrastructure is also a highly prevalent issue in low-resource settings. While long-term solutions should prioritize improving the function and availability of medical equipment and resources, our study highlights the importance of modifying existing hospital flow and healthcare costs. Dedicated health personnel to process laboratory samples, rather than reliance on patients' relatives or friends, will likely reduce the delays in obtaining crucial results that affect immediate medical management. Expanding Ghana's National Health Insurance scheme to truly cover all aspects of maternal healthcare (including purchase of medications and laboratory tests) will encourage patients to seek care while also equipping doctors with the clinical information to better analyze options for treatment.

## Strength and limitations

By using a qualitative approach and including the perspectives of a diverse sample of doctors with different clinical roles and levels of experience, this study provides a nuanced description of specific and actionable challenges to diagnosing and managing preeclampsia in urban Ghana. However, limitations still exist regarding the interpretation and generalizability of the results. The study was intentionally performed at a well-established teaching hospital and referral center for the most complicated pregnancies to ensure thoroughness and diversity of doctors' experiences. The urban setting may have caused responses to overlook differences in more rural, lower-level facilities (including fewer highly trained providers, less patient volume and acuity, and fewer health resources). Secondly, since interviews were conducted within the hospital compound and the research team included the doctors' peer obstetrics colleague, doctors could have been hesitant to share challenges about the hospital setting or their own clinical practice. However, inclusion of a local research team member was vital to the development of culturally competent research materials and appropriate sampling strategy. Moreover, interviews were intentionally performed by an objective research team member to limit this concern. Thirdly, ancillary healthcare staff such as nurses, midwives or community health workers were not included as participants. Given their critical role in obstetric care in Ghana and most LMICs as well as ability to comment on the current state of care, future research should attempt to capture their perspectives. Finally, while doctors identified challenges faced at the patient-level, they may not have direct experience or complete comprehension of the challenges patients face in accessing and adhering to recommended preeclampsia care. Therefore, future studies should also include patient perspectives.

## Conclusions

Overall, obstetric doctors in Ghana highlight that significant challenges exist to diagnosing and managing preeclampsia. Three overarching global challenges were defined as the patient-level challenge of low education levels and health literacy amongst women, the provider-level challenge of an insufficient number of highly trained obstetric healthcare providers, and the systems-level challenge of inadequate health infrastructure. Recognizing and addressing these root challenges has great potential to improve outcomes in pregnancies complicated by pre-eclampsia in low-resource settings.

## Supporting information

**S1 File. Interview guide preeclampsia challenges.**
(DOCX)

**S2 File. Standards for Reporting Qualitative Research (SRQR) checklist.**
(DOCX)

## Author Contributions

**Conceptualization:** Namratha Atluri, Titus K. Beyuo, Emma R. Lawrence.

**Data curation:** Namratha Atluri.

**Formal analysis:** Namratha Atluri, Emma R. Lawrence.

**Funding acquisition:** Namratha Atluri, Emma R. Lawrence.

**Investigation:** Namratha Atluri.

**Methodology:** Namratha Atluri, Titus K. Beyuo, Emma R. Lawrence.

**Project administration:** Namratha Atluri, Titus K. Beyuo.

**Resources:** Samuel A. Oppong, Cheryl A. Moyer.

**Supervision:** Emma R. Lawrence.

**Writing – original draft:** Namratha Atluri, Emma R. Lawrence.

**Writing – review & editing:** Titus K. Beyuo, Samuel A. Oppong, Cheryl A. Moyer.

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
