## [Decision Letter · Decision Letter 0]

22 Dec 2022

PGPH-D-22-01570

Challenges diagnosing and managing preeclampsia in a low-resource setting: A qualitative study of obstetric provider perspectives from Ghana

Dear Dr. BEYUO,

Thank you for submitting your manuscript to PLOS Global Public Health. After careful consideration, we feel that it has merit but does not fully meet PLOS Global Public Health’s publication criteria as it currently stands. Therefore, we invite you to submit a revised version of the manuscript that addresses the points raised during the review process.

The reviewers provided several methodological observations for refinement in reporting and recommendations for content revision. When returning the revised version of the manuscript to the journal, please apply the SRQR checklist (https://www.equator-network.org/reporting-guidelines/srqr/) indicating the page and paragraph numbers for the location of each reporting element.

We look forward to receiving your revised manuscript.

Kind regards,

Patrick A. Palmieri, DHSc, DPhil(Hon), EdS, MBA, MSN, PGDip(Oxon), ACNP, RN, CPHRM, CPHQ, FFNMRCSI, FAAN

Academic Editor

Journal Requirements:

2. Please send a completed 'Competing Interests' statement, including any COIs declared by your co-authors. If you have no competing interests to declare, please state "The authors have declared that no competing interests exist". Otherwise please declare all competing interests beginning with the statement "I have read the journal's policy and the authors of this manuscript have the following competing interests:"

3. Please amend your detailed Financial Disclosure statement. This is published with the article. It must therefore be completed in full sentences and contain the exact wording you wish to be published.

b. If any authors received a salary from any of your funders, please state which authors and which funders.

Additional Editor Comments (if provided): Return the SRQR checklist as noted in the decision letter.

Reviewers' comments:

Reviewer's Responses to Questions

**Comments to the Author**

1. Does this manuscript meet PLOS Global Public Health’s publication criteria? Is the manuscript technically sound, and do the data support the conclusions? The manuscript must describe methodologically and ethically rigorous research with conclusions that are appropriately drawn based on the data presented.

Reviewer #1: Yes

Reviewer #2: Yes

Reviewer #3: Yes

2. Has the statistical analysis been performed appropriately and rigorously?

Reviewer #1: Yes

Reviewer #2: N/A

Reviewer #3: N/A

3. Have the authors made all data underlying the findings in their manuscript fully available (please refer to the Data Availability Statement at the start of the manuscript PDF file)?

Reviewer #1: No

Reviewer #2: Yes

Reviewer #3: No

4. Is the manuscript presented in an intelligible fashion and written in standard English?

Reviewer #1: Yes

Reviewer #2: Yes

Reviewer #3: Yes

5. Review Comments to the Author

Reviewer #1: General Comments

A well written paper which looks at challenges in diagnosing and managing preeclampsia in Ghana from perception of health care givers especially doctors. The findings are important benchmark for improvement of maternal outcomes and managing preeclampsia in Ghana.

The study was conducted in urban Hospital and should be clearly mentioned in the discussion. I believed if similar questions were asked to health professionals in rural areas when health care to pregnancy are marginalized, the results may vary from this.

Title

1. Preposition missing in the title …... Challenges ? diagnosing and managing…

2. Can this sentence be rewritten to make more sense. The “but” in this sentence should be removed “The greatest burden of disease is in low- and middle-income countries where healthcare providers face significant, but understudied, challenges to diagnosing and managing preeclampsia”

Methods

You have stated that “Participants’ thoughts and opinions were gathered without the bias of researchers’ assumptions about preeclampsia care in Ghana.” Can you justify how this being done. Did you engage bracketing in the data collection and analysis process to ensure that any preconception notions from researchers do not taint the process ?

Discussion

You need to weave your study aims/objectives at the beginning of your discussion so that the reader knows you achieved them. This can be a sentence under “principal finding”

Strength and limitation

Is the first sentence a strength or limitation ?

Interviewing highly trained and knowledgeable clinicians. Do the researchers ever thought that this could be biased as clinicians may talked based on their knowledge rather than their experiences, such disadvantages of the population

Reviewer #2: This manuscript discusses the challenges with diagnosing and managing preeclampsia within an urban, tertiary hospital in Accra, Ghana. The study is methodologically sound and produces interesting results that contribute to the literature on this topic. With a few minor changes, I believe this manuscript is appropriate for publication.

Introduction:

The authors give a good overview of preeclampsia and the complications that result from it. It would be helpful if the authors add a paragraph about the burden of preeclampsia specifically in Ghana and its implications. Have there been any efforts to address the issue in recent years?

Methods:

The methods section is soundly written. It might be helpful to start out the section with your study design and then move into the study setting, etc. In terms of recruitment, how many total eligible participants were there? This would help explain some of the findings discussed in the results section (e.g., lack of sufficient healthcare personnel). Additionally, how was the interview guide developed? Did you use a theoretical framework or was it based on previous research? It might be helpful to include the interview guide as a supplemental file.

Results and Discussion:

The results and discussion are strong and do not require any substantial changes. I would make one remark in relation to the strengths and limitations section. The authors state that interviews were conducted by an American research team member to limit concerns with sharing challenges related to the hospital setting or clinical practice. I would caution away from framing this statement in this way and instead say the interviews were conducted by an objective member of the research team that is not affiliated with the hospital. There is a lot of hesitancy and reservations related to Americans going to LMICs and conducting research, so if you choose to include the statement as is, it would be important to also have a reflexivity statement to share your positionality.

Reviewer #3: 1. Important issue has been raised. It seems the condition of care is different between low income countries.

2. The study clearly shows the obstetric and gynecology service situation as well as religious and social life in Ghana.

3. I would recommend the researchers to take more sampling from other tertiary hospital or maternity hospitals which are providing preeclampsia and eclampsia care.

4. Or to interview the midwives to ensure the study results.

5.

6. PLOS authors have the option to publish the peer review history of their article (what does this mean?). If published, this will include your full peer review and any attached files.

**Do you want your identity to be public for this peer review?** For information about this choice, including consent withdrawal, please see our Privacy Policy.

Reviewer #1: No

Reviewer #2: No

Reviewer #3: No

---

## [Decision Letter · Decision Letter 1]

30 Jan 2023

PGPH-D-22-01570R1

Challenges diagnosing and managing preeclampsia in a low-resource setting: A qualitative study of obstetric provider perspectives from Ghana

Dear Dr. BEYUO,

Thank you for submitting your manuscript to PLOS Global Public Health. After careful consideration, we feel the manuscript requires a minor revision for publication in PLOS Global Public Health. Therefore, we invite you to submit a revised version of the manuscript that addresses the point raised about the inconsistency in the study design and the state method for data analysis. 

We look forward to receiving your revised manuscript.

Kind regards,

Patrick A. Palmieri, DHSc, DPhil(Hon), EdS, MBA, MSN, PGDip(Oxon), ACNP, RN, CPHRM, CPHQ, FFNMRCSI, FAAN

Academic Editor

Journal Requirements:

2. Please ensure that the Title in your manuscript file and the Title provided in your online submission form are the same.

Additional Editor Comments (if provided):

Please make the minor revision as requested by the reviewer.

Reviewers' comments:

Reviewer's Responses to Questions

**Comments to the Author**

1. If the authors have adequately addressed your comments raised in a previous round of review and you feel that this manuscript is now acceptable for publication, you may indicate that here to bypass the “Comments to the Author” section, enter your conflict of interest statement in the “Confidential to Editor” section, and submit your "Accept" recommendation.

Reviewer #4: (No Response)

2. Does this manuscript meet PLOS Global Public Health’s publication criteria? Is the manuscript technically sound, and do the data support the conclusions? The manuscript must describe methodologically and ethically rigorous research with conclusions that are appropriately drawn based on the data presented.

Reviewer #4: Partly

3. Has the statistical analysis been performed appropriately and rigorously?

Reviewer #4: N/A

4. Have the authors made all data underlying the findings in their manuscript fully available (please refer to the Data Availability Statement at the start of the manuscript PDF file)?

Reviewer #4: No

5. Is the manuscript presented in an intelligible fashion and written in standard English?

Reviewer #4: Yes

6. Review Comments to the Author

Reviewer #4: In the section on the research design (lines 139-142), the authors state that grounded theory was used but the explanations of data collection, analysis, and the format of the results are not consistent with grounded theory. Reference to grounded theory should be removed.

7. PLOS authors have the option to publish the peer review history of their article (what does this mean?). If published, this will include your full peer review and any attached files.

**Do you want your identity to be public for this peer review?** For information about this choice, including consent withdrawal, please see our Privacy Policy.

Reviewer #4: **Yes: **Kara L. Vander Linden, EdD

---

## [Decision Letter · Decision Letter 2]

15 Mar 2023

Challenges to diagnosing and managing preeclampsia in a low-resource setting: A qualitative study of obstetric provider perspectives from Ghana

PGPH-D-22-01570R2

Dear Dr. BEYUO,

We are pleased to inform you that your manuscript 'Challenges to diagnosing and managing preeclampsia in a low-resource setting: A qualitative study of obstetric provider perspectives from Ghana' has been provisionally accepted for publication in PLOS Global Public Health.

Best regards,

Julia Robinson

Executive Editor

Reviewer Comments (if any, and for reference):

Reviewer's Responses to Questions

**Comments to the Author**

1. If the authors have adequately addressed your comments raised in a previous round of review and you feel that this manuscript is now acceptable for publication, you may indicate that here to bypass the “Comments to the Author” section, enter your conflict of interest statement in the “Confidential to Editor” section, and submit your "Accept" recommendation.

Reviewer #4: All comments have been addressed

2. Does this manuscript meet PLOS Global Public Health’s publication criteria? Is the manuscript technically sound, and do the data support the conclusions? The manuscript must describe methodologically and ethically rigorous research with conclusions that are appropriately drawn based on the data presented.

Reviewer #4: Yes

3. Has the statistical analysis been performed appropriately and rigorously?

Reviewer #4: N/A

4. Have the authors made all data underlying the findings in their manuscript fully available (please refer to the Data Availability Statement at the start of the manuscript PDF file)?

Reviewer #4: No

5. Is the manuscript presented in an intelligible fashion and written in standard English?

Reviewer #4: Yes

6. Review Comments to the Author

Reviewer #4: (No Response)

7. PLOS authors have the option to publish the peer review history of their article (what does this mean?). If published, this will include your full peer review and any attached files.

**Do you want your identity to be public for this peer review?** For information about this choice, including consent withdrawal, please see our Privacy Policy.

Reviewer #4: **Yes: **Kara L. Vander Linden
